# Relevant Biological Effects of Varicocele Embolization with N-Butyl Cyanoacrylate Glue on Semen Parameters in Infertile Men

**DOI:** 10.3390/biomedicines9101423

**Published:** 2021-10-09

**Authors:** Olivier Chevallier, Patricia Fauque, Carole Poncelet, Kévin Guillen, Pierre-Olivier Comby, Karine Astruc, Julie Barberet, Nicolas Falvo, Emmanuel Simon, Romaric Loffroy

**Affiliations:** 1Department of Vascular and Interventional Radiology, Image-Guided Therapy Center, François-Mitterrand University Hospital, 14 Rue Paul Gaffarel, BP 77908, 21079 Dijon, France; olivier.chevallier@chu-dijon.fr (O.C.); kguillen@hotmail.fr (K.G.); nicolas.falvo@chu-dijon.fr (N.F.); 2Imaging and Artificial Vision (ImViA) Laboratory-EA 7535, University of Bourgogne/Franche-Comté, 9 Avenue Alain Savary, BP 47870, 21078 Dijon, France; pierre-olivier.comby@chu-dijon.fr; 3Laboratory of Reproductive Biology and Marital Sterility, François-Mitterrand University Hospital, 14 Rue Paul Gaffarel, BP 77908, 21079 Dijon, France; patricia.fauque@chu-dijon.fr; 4Department of Radiology, Georges-François Leclerc Cancer Center, 1 Rue du Professeur Marion, 21000 Dijon, France; cponcelet@cgfl.fr; 5Department of Neuroradiology and Emergency Radiology, François-Mitterrand University Hospital, 14 Rue Paul Gaffarel, BP 77908, 21079 Dijon, France; 6Department of Epidemiology and Biostatistics, François-Mitterrand University Hospital, 14 Rue Paul Gaffarel, BP 77908, 21079 Dijon, France; karine.astruc@chu-dijon.fr; 7Department of Gynecology, Obstetrics, and Fetal Medicine, François-Mitterrand University Hospital, 14 Rue Paul Gaffarel, BP 77908, 21079 Dijon, France; julie.barberet@chu-dijon.fr (J.B.); emmanuel.simon@chu-dijon.fr (E.S.)

**Keywords:** varicocele, sperm, infertility, oligospermia, azoospermia, embolization, glue, NBCA, lipiodol

## Abstract

Surgical treatment or varicocele embolization (VE) with sclerosing or mechanical embolic agents have been shown to improve the semen parameters of infertile men. The aim of this study was to evaluate the impact of VE using N-butyl cyanoacrylate (NBCA) glue on semen parameters in infertile men. From January 2014 to June 2018, infertile adult patients with stage 3 varicocele and an initial semen analysis showing at least one abnormal semen parameter, and who were successfully embolized with NBCA Glubran^®^2 glue, were retrospectively recruited. The availability of a second semen analysis after VE was mandatory for patient inclusion. The primary endpoint was the change in total sperm number (TSN) after VE. The other parameters of interest were progressive and total sperm motilities (Smot) at 1 h (H1), sperm vitality (SV) and morphology (SMor). One hundred and two patients were included. Eight patients presented null TSN before and after VE. Among the remaining 94 patients, a significant improvement in the median TSN after VE was shown (31.79 × 10^6^/ejaculate [IQR: 11.10–127.40 × 10^6^/ejaculate] versus 62.24 × 10^6^/ejaculate [IQR: 17.90–201.60 × 10^6^/ejaculate], *p* = 0.0295). Significant improvement in TSN was found for the 60 oligo- or azoospermic patients (*p* = 0.0007), whereas no significant change was found for the 42 patients with normal initial TSN (*p* = 0.49). Other parameters, such as progressive and total SMot, SV and SMor, also significantly improved after VE (*p* = 0.0003, 0.0013, 0.0356 and 0.007, respectively). The use of NBCA glue as an embolic agent for VE in infertile men with stage 3 varicocele significantly improves the semen parameters.

## 1. Introduction

Varicocele is caused by a reflux of blood in the internal spermatic vein and is defined by abnormal tortuosity and dilatation of the veins in the pampiniform plexus. It can lead to the failure of ipsilateral testicular growth and development, symptoms of pain and discomfort and reduced fertility [1]. The frequency of varicocele can be as high as 22% in the general population [2]. Varicocele is found in 40% of men with an abnormal semen analysis [2]. The etiology of varicocele is debated and probably multifactorial. Congenital and/or acquired valve dysfunction responsible for reflux in the internal spermatic vein are the most common causes [3]. The left side is affected in 80–90% of cases whereas varicocele is present on the right side in only 5–10% of cases, the condition being bilateral in 1–15% [3]. The link between varicocele repair and improvement in sperm parameters was first described by Tulloch et al. in 1955 [4]. Surgical varicocelectomy through ligature has long been regarded as the reference treatment, and has demonstrated good results for improving the semen parameters of infertile men with varicoceles [5,6,7]. In 1987, Sigmund et al. described the percutaneous transvenous embolization [8], which showed good short- and long-term results [9,10]. Varicocele embolization (VE) has proven to be just as effective as surgical ligation in improving male infertility with the advantage of shorter recovery to full activity as compared to surgical ligation [11]. VE can also be performed after surgical failure since this method has the ability to detect gonadal vein variants [12]. Sclerosing agents, coils and balloons have been widely used as percutaneous embolic agents [9,10,11,13,14,15,16]. The use of biological glues in varicocele embolization was initiated in 1980 by Kunnen et al. using isobutyl-2-cyanoacrylate (IBCA, Bucrylate^®^, Ethicon, Raritan, NJ, USA) [17]. This glue was later replaced with n-butyl-2-cyanoacrylate (NBCA or Enbucrilate; Histoacryl^®^, B. Braun, Tuttlingen, Germany) for reasons of possible carcinogenicity [18]. Studies have reported the use of glues for VE with very good results [19,20,21,22,23]. However, embolization with cyanoacrylate remains challenging, as the occlusion is permanent and virtually instantaneous [24]. Glubran^®^2 (GEM SRL, Viareggio, Italy) is a recently developed surgical glue in which N-butyl 2-cyanoacrylate is combined with another monomer, metacryloxysulpholane (NBCA-MS), to produce a more pliable polymer whose milder exothermic reaction (45 °C) results in less inflammation and histotoxicity [25]. NBCA and NBCA-MS have been found to be equally efficient and safe for the embolization of varicoceles [20]. Some studies found no difference in pain experience with NBCA-MS as compared to NBCA [21,26]. In addition, less radiation and lower recurrence rates were found with the use of Glubran^®^2 for VE than with the use of mechanical agents (coils and/or plugs) and a sclerosing agent (polidocanol) [23]. Embolization with NBCA-MS is an uncomplicated, inexpensive, efficient, and safe technique [19]. However, to our knowledge, no study has yet assessed specifically the changes in semen parameters after VE using NBCA-MS in infertile men. The aim of our study was to evaluate the impact of VE using NBCA-MS glue as an embolic agent on semen parameters in infertile men.

## 2. Materials and Methods

### 2.1. Patients

In this retrospective study, all VE procedures performed from January 2014 to June 2018 at our tertiary institution in patients with stage 3 varicocele and couple infertility, and an initial semen analysis showing at least one abnormal semen parameter according to the 2010 WHO criteria, were identified by searching the hospital’s electronic database. Infertility was characterized by the failure to establish a clinical pregnancy after 12 months of regular, unprotected sexual intercourse or due to an impairment of a person’s capacity to reproduce either as an individual or with his/her partner [27].

Inclusion criteria were as follows: successful VE using NBCA-MS Glubran^®^2 confirmed by a control venography at the end of the procedure and by a scrotal ultrasonography 3–6 months after embolization, or later if there was an oversight. A second semen analysis was available to compare the semen parameters before and after VE. Exclusion criteria were as follows: patients < 18 years of age, patients with no infertility, patients for whom VE was performed only for pain or with an embolic agent other than Glubran^®^2, and patients without second semen analysis performed after VE. Numerous patients had no second semen analysis and the reason was retrospectively ascertained by phone.

The following data were also recorded: age, body mass index (BMI), exposure to heat (exposed trades: baker, cook, mason, truck driver), exposure to toxic substances (exposed trades: farming, winemaker, painter), tobacco consumption (active, cessation, non-existent), type of infertility (primary or secondary), the size of embolized varicoceles.

A Data Institutional Review Board was not required for this study due to its retrospective nature. Given that VE was a common practice, the local ethics committee did not request additional consent.

### 2.2. Varicocele Diagnosis

The diagnoses of varicocele and gonadal vein insufficiency were confirmed in all patients by physical examination and high-resolution scrotal ultrasonography (US) with color-flow Doppler analysis using the Aixplorer Imagine ultrasound system (Supersonic Imagine, Aix-en-Provence, France). The varicoceles were classified according to an adapted version of the Dubin and Amelar classification as subclinical grade 1 (small), grade 2 (moderate), and grade 3 (large) [28]. Battino’s classification was used for the Doppler study, which takes into account the duration of reflux [29]:Stage 1: moderate spontaneous hyperflow, frank reflux the duration of which is less than the time of thrust but longer than two seconds;Stage 2: significant spontaneous hyperflow, frank reflux lasting as long as the push;Stage 3: strong spontaneous hyperflow, massive reflux not only at the Valsalva maneuver but also at each breath in.

Only stage 3 varicoceles were treated by VE. During the US exam, the volume of each testicle was measured and the side of varicocele was noted. The testicular volume was defined as 0.71 × length × width × height. Normal testicular volume and testicular hypotrophy were defined as a volume greater than 15 mL and less than 10 mL, respectively. Volumes between 10 and 15 mL were considered borderline. The normal total (right plus left) testicular volume was defined as 32 mL or more [30].

A second scrotal US was performed generally 3 to 6 months after VE, and the treatment was considered successful when no reflux in the pampiniform plexus was observed.

### 2.3. Semen Analyses

Sperm analyses were performed in the Reproductive Biology unit of our institution. Samples were obtained by masturbation after 2–5 days of sexual abstinence. After liquefaction, standard semen parameters (total sperm number, motility, vitality, morphology) were measured according to the 2010 WHO guidelines (fifth edition) [31]. Semen analyses were performed prior to VE (M0) and then after the interventional treatment. The values considered normal were as follows:Total sperm number (TSN): ≥39 × 10^6^/ejaculate;Sperm motility (SMot): progressive motility (grades A+B): >32% at 1 h (H1), total motility (grades A+B+C): >40% at 1 h (H1);Sperm vitality (SV): ≥58%;Sperm morphology (SMor): ≥23%.

### 2.4. Embolization Technique

The goal of embolization was to occlude the spermatic vein along with the parallel and segmental veins, covering the origins of all relevant side branches or renospermatic bypasses. All procedures were performed in an outpatient setting by one experienced interventional radiologist (R.L.), using a Philips Allura Xper FD 20 angio room (Philips, Best, The Netherlands). Intravenous sedation and local anesthesia were generally provided for patient comfort. The right femoral vein was usually accessed under US guidance. In conjunction with a hydrophilic guide wire, a 5-French Cobra catheter was used to catheterize the left renal vein and then the orifice of the left gonadal vein. Right jugular vein approach was used in cases where the femoral approach resulted in unsuccessful catheterization of the left or right gonadal vein. This approach takes advantage of the less-acute angle with the right gonadal vein origin since it is usually located on the right anterolateral inferior vena cava just below the right renal vein. After catheterizing the gonadal vein, pre-embolization venography was performed under Valsalva maneuver. A 2.7-Fr microcatheter (Progreat^®^, Terumo Interventional Systems, Japan) was then used to catheterize the gonadal vein. Venograms were obtained at various levels while advancing the microcatheter from the orifice of the gonadal vein to the level of the pubic symphysis. The paraspermatic veins, connecting collaterals or renospermatic bypasses to the internal spermatic vein, were mapped. In our study, the embolization procedure was performed using 1 to 3 mL of Glubran^®^2 mixed with Lipiodol^®^ (Lipiodol^®^ Ultrafluid, Guerbet, Aulnay-sous-Bois, France) at a ratio of 1:1 to get fast polymerization and avoid migration in the case of reflux. The microcatheter dead space was first filled with an anionic solution, dextrose 5%, to avoid intracatheter glue polymerization. Glue–Lipiodol^®^ mixture was injected under strict fluoroscopic control, with continuous manual injection and a display of real-time distribution. The glue–Lipiodol^®^ mixture injection began in the distal intrapelvic segment of the gonadal vein, and the microcatheter was withdrawn while injecting the glue under fluoroscopy guidance. Injection was stopped before the pampiniform plexus was reached. The microcatheter was removed when the venous space selected beforehand was filled by the glue–Lipiodol^®^ mixture [23,32,33,34].

All patients were discharged after 2 h of observation and returned to their normal daily activities after 24 h. Patients were asked to avoid heavy physical activity for 7 days and Piroxicam 20 mg twice daily was prescribed for one week.

Figure 1 shows an example of bilateral VE procedure using Glubran^®^2.

### 2.5. Outcomes

The primary endpoint was the change in TSN after VE. Effect from the patients’ characteristics on the variation in the TSN was also assessed. Secondary outcomes were changes from the mean baseline of each semen parameter (SMot, SV, SMor). Minor and major complications were also reported.

### 2.6. Statistical Analyses

Categorial variables were presented as percentages and compared using a Chi2 test. Continuous variables were described as means with standard deviations (SD) and medians with interquartile ranges (IQR) values. They were compared using parametric tests (Student or ANOVA). Paired student *t* test was used to compare continuous variables before and after varicocele embolization. Statistical analyses were performed using Stata 14.0 software (StataCorp., College Station, TX, USA).

## 3. Results

### 3.1. Study Population

During the study period, 186 infertile patients with stage 3 varicocele underwent VE. Among these 186 patients, 159 VE were performed using Glubran^®^2 as an embolic agent with 100% technical success, and a second semen analysis after VE was available for 102 of the 186 patients. Among the 159 patients who were treated with Glubran^®^2, a second semen analysis was not obtained for 57 of them. The reason for the absence of this exam was retrospectively asked by phone interview. Twenty-eight patients answered and gave the following reasons: eleven patients stated that the exam was not prescribed, seven had wife pregnancy/childbirth shortly after VE and did not find useful to do the exam, two underwent the second semen analysis in another city, four forgot to do this exam, three stopped medically assisted procreation follow-up, and one refused this exam. Figure 2 shows a flow chart of the patients’ selection. Finally, 102 patients with a mean age of 32.3 years were included in this retrospective study. The patients’ characteristics are summarized in Table 1. Mean BMI was 24.96 kg/m^2^. The majority of patients did not present heat (59.1%) or toxic exposures (78.5%) and never smoked (54.1%). The infertility was primary for 85.3% of patients. The most frequently embolized side was the left one (90.2%). The median delays between the first semen samples and VE and between VE and second semen samples were, respectively, 87 days (IQR: 54–161) and 97 days (IQR: 91–134).

### 3.2. Total Sperm Number

Eight patients presented a null total sperm number before and after VE. Among the remaining 94 patients, a significant improvement in the numeration after VE was observed as compared to baseline (*p* = 0.0295). Before and after VE, the median TSN were 31.79 × 10^6^/ejaculate (IQR: 11.10–127.40) and 62.24 × 10^6^/ejaculate (IQR: 17.90–201.60), respectively (Table 2). Among these 94 patients, a greater improvement of the TNS was found in the 52 patients who presented initial abnormal TSN than in the 42 with normal initial TSN (median, 133% [IQR: −6–648] versus 4% [IQR: −45–104]). Box plots shown in Figure 3 confirm these results with a significant improvement in the TSN among initially oligo or azoospermic patients (*p* = 0.0007), whereas there was no significant change for patients with normal initial TSN (*p* = 0.49). Cross-tabulation analysis shows that 19 patients with abnormal initial TSN obtained a normal TSN after VE, while six patients with an initial normal TSN presented an abnormal TSN after VE (Table 3). In addition, Table 4 describes the impact of patients’ characteristics on the TSN changes. There was no significant difference for TSN between the different groups before and after VE for any of the criteria: heat exposure, toxic exposure, tobacco consumption, type of infertility, age and BMI. In addition, there was no significant improvement in TSN after VE for any of the criteria studied, whether patients were in one group or another.

### 3.3. Sperm Motility, Vitality and Morphology

Data on SMot, SV and SMor were available before and after VE for, respectively, 92, 90 and 64 patients. After VE, significant improvements in the other semen parameters, progressive and total SMot, SV and SMor, were also found (*p* = 0.0003, 0.0013, 0.0356, and 0.0007, respectively) (Table 2). Cross-tabulation analyses also highlighted improvements in progressive and total SMot, SV and SMor (Table 5, Table 6, Table 7 and Table 8, respectively) after VE. Forty-eight patients (52.2%) showed abnormal initial progressive SMot against 28 patients (30.4%) on the second analysis. Similarly, an improvement in total motility was found in 50 patients (54.4%) who presented an initial normal total Smot as compared to 67 (72.8%) after VE. Among 45 patients (50.0%) with initially abnormal SV, 24 (26.7%) demonstrated normal values after VE. While only seven patients (10.9%) presented initial normal SMor, SMor was considered normal for 17 patients (26.6%) after VE.

### 3.4. Testicular Volume

Data concerning testicular volume before VE were available for 102 patients, whereas these data were available for 60 patients before and after VE. Among 95 patients who underwent unilateral VE, before VE, a significant difference in testicular volume was found between the testicle on the varicocele side and the contralateral testicle (mean volumes 12.69 ± 4.04 mL *versus* 14.85 ± 4.66 mL, *p* = 0.00001). Out of the 60 patients, seven underwent bilateral embolization and 53 patients underwent unilateral embolization. There was no significant difference in testicular volumes before and after VE for either embolized or contralateral testicles (*p* = 0.251 and *p* = 0.630, respectively) (Table 9).

### 3.5. Complications

No major complications were reported during or after VE at mean clinical and radiological follow-up evaluations. No phlebitis of the pampiniform plexus was observed in the all-study population. No glue migration into the lung or nontarget vessels was reported during or after VE. Post-embolization syndrome, including groin discomfort on the embolized side within 1 month, was observed in 26 (14%) of the 186 patients, which was spontaneously resolutive with conservative management. No early recurrence of varicocele was noted at mean US follow-up.

## 4. Discussion

With 102 subjects included, this study demonstrated that VE using NCA-MS Glubran^®^2 glue as an embolic agent significantly improves the spermogram parameters in infertile patients with stage 3 varicocele. Significant improvements were found for all parameters that were assessed: the TSN (*p* = 0.0295), the progressive SMot at H1 (*p* = 0.0003), the total SMot at H1 (*p* = 0.0013), the SV (*p* = 0.0356) and SMor (*p* = 0.0007). Significant improvement in the TSN was present for patients who were initially oligo or azoospermic (*p* = 0.0007), whereas there were no significant changes for patients with normal initial TSN (*p* = 0.49).

Our results are consistent with literature reports. Two meta-analyses demonstrated that surgical varicocele repair significantly improved sperm parameters [6,7]. VE has already been shown to improve sperm parameters using embolic agents other than glue, such as sclerotherapy and mechanical agents (such as coils and plugs) [9,35,36,37]. In Riedl et al.’s study, improvement in sperm parameters (sperm count, sperm motility and differential spermogram) after VE was defined as an increase of at least 50% of the initial value, and was found in 25/32 subjects (78.1%) for at least one parameter of viability, and in 4/32 (78%) for all three parameters, the sperm count being the parameter that was the most frequently improved (62.5%) [9]. In Galfano et al.’s study, in 188 patients with low sperm number, sperm count significantly improved from 12 to 19.5 × 10^6^/mL in 336 patients with asthenospermia, progressive motile forms significantly improved from 25% to 45%, and in 147 patients with teratospermia, normal forms increased from 17% to 35% [35]. In 206 patients with complete or partial improvement of incontinence degree after VE, significant improvements in sperm concentration (*p* = 0.002) and progressive motility (*p* = 0.0001) were found in Di Bisceglie et al.’s study, whereas no improvement was found for the sperm morphology, contrary to our study results [36]. With 47 infertile patients with at least one abnormal semen parameter, significantly higher values of sperm concentration, sperm progressive motility, vitality and percentage of normal sperm were found after VE in Prasivoravong et al.’s study [37]. To our knowledge, no previous study assessed the effect of VE using NBCA on sperm parameters.

Few studies reported the use of glues for VE, with very high technical success [19,20,21,22,23]. The main advantage of cyanoacrylate is the lasting nature of the vascular occlusion [24,38]. It is also an effective, safe, inexpensive embolic agent for VE [23]. Glue is also effective in the presence of collateral vessels. Due to its liquid nature, the glue can expand into collateral veins allowing the occlusion of these vessels in addition to the main gonadal vein. The use of NBCA glue allows for the effective treatment of persistent or recurrent varicoceles after surgical repair that are usually due to anatomical variants of the gonadal vein [22]. The primary disadvantage of liquid embolic agents compared to coils or nitinol plugs is that their release is not 100% controllable. Potential complications of cyanoacrylate VE include glue migration into the pulmonary circulation, a glued catheter, and severe phlebitis of the gonadal vein or pampiniform plexus, although this is exceptional [39]. The glue should be injected slowly but decisively to avoid reflux of the material leading to occlusion of non-target vessels.

Glubran^®^2 is a recently developed surgical glue in which NBCA is combined with another monomer, metacryloxysulpholane (NBCA-MS), to produce a more pliable polymer whose milder exothermic reaction (45 °C) results in less inflammation and histotoxicity. The preparation must be sufficiently fluid. Polymerization occurs nearly instantaneously. It is a non-toxic biocompatible liquid embolic agent with a density analogous to water. It is transparent, colorless, and has a typical odor. It presents as being equally as efficient and safe as NBCA [20], with the same level of pain [21,26]. Embolization with Glubran^®^2 showed high technical success with rare complications [19]. In addition, the use of Glubran^®^2 for VE has been shown to lead to less radiation and lower recurrence rates than with other embolic materials, such as mechanical agents and sclerosing agents. In addition, Glubran^®^2 benefits from Community European (CE) marking, which is not the case for Histoacryl^®^, the use of this last glue being thus theoretically not allowed for endovascular purposes and is off-label.

Most of the patients suffered from primary infertility in our study. However, no significant difference was found regarding the incidence of varicoceles in men with primary and secondary infertility in a previous report [40]. Deleterious cofactors of fertility have been described by Nevoux et al. [41]. Some cofactors can exacerbate the oxidative stress already induced by the presence of varicocele. At the forefront of these cofactors, smoking is the most incriminated [41]. Dorfman et al. showed a possible reversibility of the reduced fecundity associated with smoking within 1 year of cessation [42]. In our study, there was no significant change in the TSN after VE in obese patients (BMI ≥ 30 kg/m^2^), nor in patients who were smokers.

In our study, testicular volume on the varicocele side was significantly lower than that on the other side, which is consistent with previously published results [43]. No significant difference in the testicular volume before and after VE for either embolized or non-embolized testicles was found in our study. This is in contrast with the results from Sakomato et al.’s study, in which an improvement in the volume of the left testicle was found after varicocele repair in adult patients with clinical left varicocele [44]. Sakomato et al. also showed that left clinical testicular varicocele was associated with relative ipsilateral testicular hypotrophy in infertile patients [45]. However, a subclinical varicocele was not important in determining differences between right and left testicular volumes [45].

Our study had several limitations. First, this was a retrospective review of a cohort from a single center. Second, follow-up scrotal ultrasonography was not performed for all patients and/or data were missing. However, no reflux was observed after VE for all patients who underwent follow-up scrotal imaging. Third, pregnancy rates after VE were not assessed. In a review study, although authors found evidence suggesting that treatment of a varicocele in men from couples with otherwise unexplained subfertility might improve a couple’s chance of pregnancy, the findings remained inconclusive as the quality of the available evidence was very low [46]. However, in another longitudinal study, 37.4% of infertile patients without persistent varicocele after VE fathered a child [35]. In addition, the pregnancy rate also depends on parameters related to the woman. The fate of these patients and the outcome of their fertility project could be the subject of another study. Fourth, the study population of the present study was small, although VE is relatively common. However, VE often takes place in individuals with testicular pain and children with palpable varicocele, without fertility problems [14,23]. Our population here was very selected and homogeneous. In addition, in our center, VE in infertile men has sometimes been performed with embolic agents other than Glubran^®^2, such as mechanical or sclerosing agents, at least at the beginning of our experience. Fifth, due to the fact that no systematic hormonal test was performed, hypogonadism was potentially missed in some patients with oligo- or azoospermia who may not have required VE. Lastly, parameters of the spermogram are subject to intra- and inter-individual variabilities [47]. It may be advisable to perform several semen analyses before and after VE to reduce these variables.

## 5. Conclusions

In conclusion, the use of NBCA-MS Glubran^®^2 glue as an embolic agent for percutaneous VE in infertile men with stage 3 varicocele improves the total sperm number and the other semen parameters such as sperm motility (progressive and total), vitality and morphology, as has been shown with mechanical agents (coils and/or plugs), sclerosing agents and varicocelectomy. Embolization with Glubran^®^2 has the advantages of being a safe, fast and cost-effective procedure. Randomized trials comparing the main embolic agents, their effects on the semen parameters and pregnancy rates and their overall characteristics are warranted.

## Figures and Tables

**Figure 1 biomedicines-09-01423-f001:**
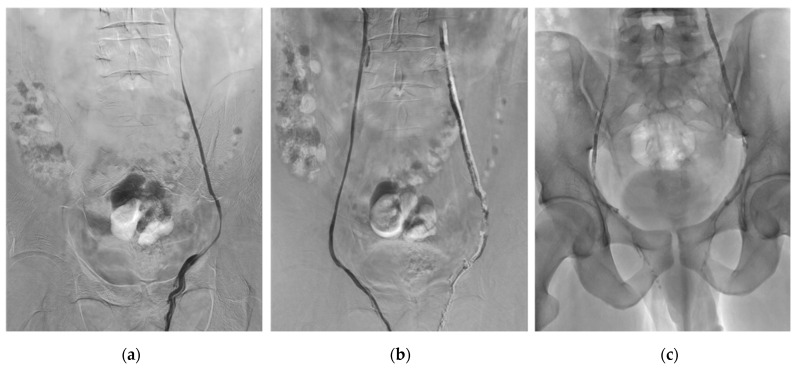
Example of typical bilateral varicocele embolization (VE) procedure using NBCA-MS Glubran^®^2. (**a**) Left gonadal vein venography through a 5-French Cobra catheter by transvenous femoral approach. (**b**) Right gonadal vein venography after VE of the left gonadal vein using Glubran^®^2-Lipiodol^®^ mixture at a 1:1 ratio. (**c**) Plain X-ray image after bilateral VE using Glubran^®^2-Lipiodol^®^ mixture at a 1:1 ratio. In more detail, a microcatheter is placed below the iliopectinal line and the dead space is filled with 5% dextrose solution to avoid intracatheter glue polymerization. The embolization procedure is then performed using glue mixed with Lipiodol at a ratio of 1:1. Glue–Lipiodol mixture is then injected under strict fluoroscopic guidance, with continuous injection performed manually and a display of real-time distribution. The glue injection begins in the distal intrapelvic segment of the gonadal vein, and the catheter is withdrawn slowly while injecting NBCA-MS under fluoroscopy. Injection is then stopped before the pampiniform plexus is reached. The microcatheter is then removed when the glue fills the venous space selected beforehand. Here we can see the cast of glue along the left and right gonadal veins after embolization.

**Figure 2 biomedicines-09-01423-f002:**
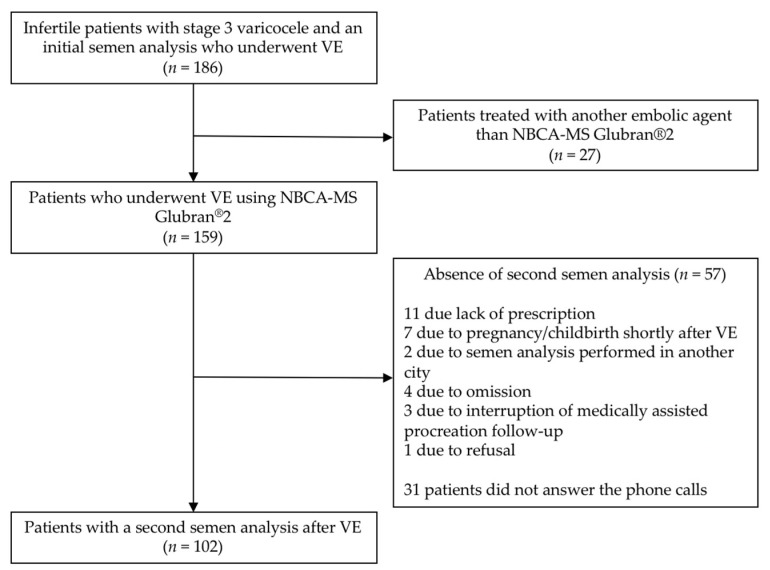
Flow chart of patients’ selection. VE, varicocele embolization; NBCA-MS, n-butyl cyanoacrylate metacryloxysulfolane.

**Figure 3 biomedicines-09-01423-f003:**
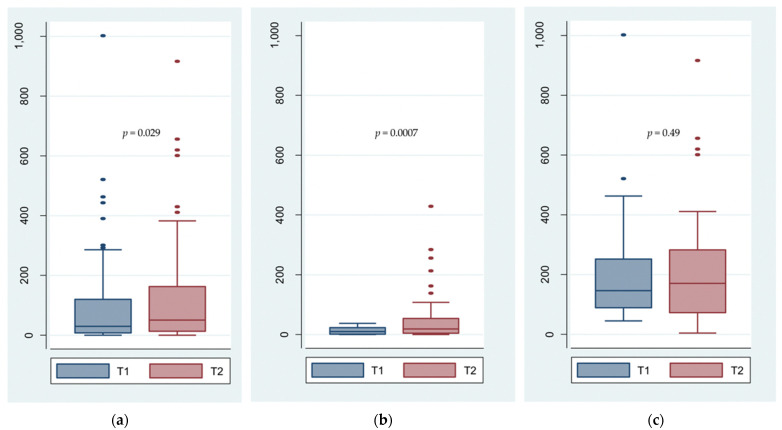
Box plots showing the total sperm number (TSN, 10^6^/ejaculate) before (T1) and after (T2) varicocele embolization with Glubran^®^2. (**a**) Evolution of TSN in all patients (*n* = 102) between T1 and T2 showing a significant improvement (*p* = 0.029); (**b**) Evolution of TSN in patients with initial abnormal TSN (*n* = 60) showing a significant improvement (*p* = 0.0007); (**c**) Evolution of TSN in patients with normal initial TSN (*n* = 42) with no significant change between T1 and T2 (*p* = 0.49).

**Table 1 biomedicines-09-01423-t001:** Patients characteristics.

Characteristics	No. of Patients *	No. (%) **
Age (years)	102	
Mean (95% CI)		32.3 (31.3–33.3)
Median (IQR)		32 (29–35)
BMI (kg/m^2^)	97	
Mean (95% CI)		25 (24.1–25.9)
Median (IQR)		24.8 (22.2–27.0)
Heat exposure	93	
No		55 (59.1)
Yes		38 (40.9)
Toxic exposure	93	
No		73 (78.5)
Yes		20 (21.5)
Tobacco consumption	98	
Never		53 (54.1)
Active		30 (30.6)
Cessation		15 (15.3)
Infertility type	102	
Primary		87 (85.3)
Secondary		15 (14.7)
Sperm alteration type	102	
Normal TSN		42 (41.2)
Oligospermia		52 (51.0)
Azoospermia		8 (7.8)
Embolized side	102	
Left		92 (90.2)
Right		3 (2.9)
Both		7 (6.9)

No., number; IQR, interquartile range; CI, confidence interval; BMI, Body Mass Index; TSN, total sperm number; *, Number of patients with available data; **, Number of patients (percentage) unless otherwise specified.

**Table 2 biomedicines-09-01423-t002:** Comparison of semen parameters before and after varicocele embolization (VE) with Glubran^®^2.

Semen Parameters	No.	Before VE	After VE	*p* Value *
Total sperm number (×10^6^/ejaculate)	94			0.0295
Mean ± SD		95.35 ± 146.24	126.24 ± 165.58	
Median (IQR)		31.79 (11.10–127.40)	62.24 (17.90–201.60)	
Progressive sperm motility at H1 (%)	92			0.0003
Mean ± SD		31.49 ± 15.24	38.02 ± 14.95	
Median (IQR)		30.5 (20.0–43.5)	38.0 (29.0–48.0)	
Total sperm motility at H1 (%)	92			0.0013
Mean ± SD		40.50 ± 14.67	46.14 ± 13.79	
Median (IQR)		41.0 (31.0–53.0)	47.0 (38.0–55.0)	
Sperm vitality (%)	90			0.0356
Mean ± SD		57.87 ± 15.30	62.18 ± 15.20	
Median (IQR)		57.0 (50.0–70.0)	62.5 (52.0–72.0)	
Sperm morphology (%)	64			0.0007
Mean ± SD		9.42 ± 8.73	14.22 ± 9.83	
Median (IQR)		7.0 (3.0–11.5)	13.0 (6.0–23.0)	

No., number of patients with available data; SD, standard deviation; IQR, interquartile range; H1, at 1 h; VE, varicocele embolization; *, test student paired data.

**Table 3 biomedicines-09-01423-t003:** Cross-tabulation table showing the number of patients with abnormal and normal total sperm number (TSN) on the first semen analysis before varicocele embolization (VE) *versus* those on the second semen analysis after VE.

		After VE		
	Total Sperm Number	Abnormal, No. (%)	Normal, No. (%)	Total, No. (%)
**Before VE**	Abnormal, No. (%)	33 (35.1)	19 (20.2)	52 (55.3)
	Normal, No. (%)	6 (6.4)	36 (38.3)	42 (44.7)
	Total, No. (%)	39 (41.5)	55 (58.5)	94 (100.0)

VE, varicocele embolization; No., number of patients.

**Table 4 biomedicines-09-01423-t004:** Impact of patients’ characteristics on the total sperm number (TSN) (10^6^/ejaculate) before and after varicocele embolization (VE).

Characteristics	No. of Patients	Before VE	*p* Value	After VE	*p* Value	Comparison Test of Percentage Variation (*p* Value)
		Mean ± SD Median (IQR)		Mean ± SD Median (IQR)		
Heat exposure						
No	55	95.54 ± 160.61	0.356	113.53 ± 146.30	0.794	0.361
		35.00 (8.42–120.06)		62.08 (13.26–162.77)		
Yes	38	67.95 ± 105.88		104.74 ± 176.28		
		15.81 (3.87–89.4)		27.27 (8.04–138.6)		
Toxic exposure						
No	73	88.62 ± 153.46	0.572	115.99 ± 169.90	0.484	0.341
		28.49 (5.43–108.00)		50.12 (13.37–138.60)		
Yes	20	68.38 ± 80.36		87.84 ± 106.70		
		36.34 (10.90–99.03)		31.00 (5.82–182.19)		
Tobacco consumption						
No	68	88.46 ± 149.88	0.900	113.56 ± 144.46	0.948	0.667
		29.85 (7.31–127.23)		53.93 (13.19–171)		
Yes	30	84.46 ± 134.14		111.27 ± 185.36		
		27.09 (4.35–97.50)		31.25 (9.13–162.77)		
Infertility						
Primary	87	92.02 ± 150.65	0.482	117.73 ± 159.69	0.847	0.636
		30.12 (8.42–120.06)		62.08 (12.60–182.00)		
Secondary	15	63.81 ± 81.91		108.96 ± 183.43		
		23.10 (1.64–97.50)		36.89 (20.72–81.60)		
Age						
≤35 years	78	95.94 ± 157.03	0.306	133.31 ± 178.35	0.058	0.337
		29.85 (9.24–120.06)		53.93 (15.96–212.42)		
>35 years	24	61.67 ± 76.78		61.62 ± 72.69		
		32.20 (3.90–85.29)		44.40 (5.82–84.50)		
Body Mass Index						
<30 kg/m^2^	84	75.20 ± 106.82	0.052	112.14 ±155.88	0.566	0.880
		29.10 (7.00–106.90)		50.56 (13.25–161.39)		
≥30 kg/m^2^	18	147.01 ± 246.45		136.51 ± 193.95		
		42.18 (7.70–175.68)		46.53 (8.04–218.30)		

No., number of patients; VE, varicocele embolization; SD, standard deviation; IQR, interquartile range.

**Table 5 biomedicines-09-01423-t005:** Cross-tabulation table showing the number of patients with abnormal and normal progressive sperm motility at 1 h (H1) in the first semen analysis before varicocele embolization (VE) *versus* in the second semen analysis after VE.

		After VE		
	Progressive Motility *	Abnormal, No. (%)	Normal, No. (%)	Total, No. (%)
**Before VE**	Abnormal, No. (%)	21 (22.8)	27 (29.4)	48 (52.2)
	Normal, No. (%)	7 (7.6)	37 (40.2)	44 (47.8)
	Total, No. (%)	28 (30.4)	64 (69.6)	92 (100.0)

VE, varicocele embolization; No., number of patients; *, progressive sperm motility (grades A+B) at H1.

**Table 6 biomedicines-09-01423-t006:** Cross-tabulation table showing the number of patients with abnormal and normal total sperm motility at 1 h (H1) in the first semen analysis before varicocele embolization (VE) *versus* in the second semen analysis after VE.

		After VE		
	Total Motility *	Abnormal, No. (%)	Normal, No. (%)	Total, No. (%)
**Before VE**	Abnormal, No. (%)	16 (17.4)	26 (28.2)	42 (45.6)
	Normal, No. (%)	9 (9.8)	41 (44.6)	50 (54.4)
	Total, No. (%)	25 (27.2)	67 (72.8)	92 (100.0)

VE, varicocele embolization; No., number of patients; *, total sperm motility (grades A+B+C) at H1.

**Table 7 biomedicines-09-01423-t007:** Cross-tabulation table showing the number of patients with abnormal and normal sperm vitality in the first semen analysis before varicocele embolization (VE) *versus* in the second semen analysis after VE.

		After VE		
	Sperm Vitality	Abnormal, No. (%)	Normal, No. (%)	Total, No. (%)
**Before VE**	Abnormal, No. (%)	21 (23.3)	24 (26.7)	45 (50.0)
	Normal, No. (%)	12 (13.3)	33 (36.7)	45 (50.0)
	Total, No. (%)	33 (36.6)	57 (63.4)	90 (100.0)

VE, varicocele embolization, No., number of patients.

**Table 8 biomedicines-09-01423-t008:** Cross-tabulation table showing the number of patients with abnormal and normal sperm morphology in the first semen analysis before varicocele embolization (VE) *versus* in the second semen analysis after VE.

		After VE		
	Sperm Morphology	Abnormal, No. (%)	Normal, No. (%)	Total, No. (%)
**Before VE**	Abnormal, No. (%)	44 (68.8)	13 (20.3)	57 (89.1)
	Normal, No. (%)	3 (4.7)	4 (6.3)	7 (10.9)
	Total, No. (%)	47 (73.4)	17 (26.6)	64 (100.0)

VE, varicocele embolization, No., number of patients.

**Table 9 biomedicines-09-01423-t009:** Testicular volumes (mL) before and after varicocele embolization (VE).

	Before VE	After VE	*p* Value
Embolized side (*n* = 60) *			
Mean volume ± SD (mL)	12.30 ± 3.88	12.64 ± 4.16	0.251
p50 (IQR: p25–p75)	12.00 (10.05–15.35)	13.25 (10.00–15.35)	
Non-embolized side (*n* = 53) *			
Mean volume ± SD (mL)	14.62 ± 4.45	14.79 ± 4.27	0.630
p50 (IQR: p25–p75)	15.00 (12.00–17.00)	15.00 (12.30–18.00)	

VE, varicocele embolization; IQR, interquartile range; n, number; *, data concerning testicular volumes were available for 60 patients before and after VE; 7 patients underwent bilateral VE.

## Data Availability

All the study data are reported in this article.

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
