# Peer review of "Relevant Biological Effects of Varicocele Embolization with N-Butyl Cyanoacrylate Glue on Semen Parameters in Infertile Men"

_biomedicines, 2021, doi:10.3390/biomedicines9101423_

Round 1

Reviewer 1 Report

  • Summary

       This manuscript addresses the issue regarding usefulness of varicocele embolization with glue. The authors evaluated semen parameters in 109 patients who underwent N-butyl cyanoacrylate (NBCA) embolization for the treatment of varicocele. They claimed that the embolization is effective in infertile men with stage 3 varicocele to improve the semen parameters.

    Evaluation

      In my opinion, the manuscript is well written and a practically and scientifically useful article. They had no complications in NBCA embolization of the gonadal vein. The authors described the embolization technique in detail. I agree with the tip of the technique which is infusion of 50%NBCA・Lipiodol mixture into a large 2.7F coaxial catheter.  

    I support to accept the article in the journal.

Author Response

Responses to Reviewer 1 Comments

Thank you very much for your relevant comments. Please find below our replies.

Summary

This manuscript addresses the issue regarding usefulness of varicocele embolization with glue. The authors evaluated semen parameters in 109 patients who underwent N-butyl cyanoacrylate (NBCA) embolization for the treatment of varicocele. They claimed that the embolization is effective in infertile men with stage 3 varicocele to improve the semen parameters.

Reply : Thank you very much for your comments.

Evaluation

In my opinion, the manuscript is well written and a practically and scientifically useful article. They had no complications in NBCA embolization of the gonadal vein. The authors described the embolization technique in detail. I agree with the tip of the technique which is infusion of 50% NBCA・Lipiodol mixture into a large 2.7F coaxial catheter.  

Reply : Thank you very much for your comments.

I support to accept the article in the journal.

Reply : Thank you very much for your comments.

Reviewer 2 Report

this is an interesting paper.  it is not clear if the first surgical indication was the infertility or the abnormal semen analysis. another question is the semen improvement after surgery still remain under "normal" values or had similar abnormalities. 

Again,had  those having a pregnancy immediatly after surgery, a severe abnormal semen before surgery? 

probably it is interesting to undestand who had the best improvement indipendently by the procedure used!  

due to the fact that there are no hormonal test it is important to explain during your limits that some of these patients may had hypogonadism, and sometime azoospermia or oligo may not required suregry

Author Response

Responses to Reviewer 1 Comments

Thank you very much for your relevant comments. Please find below our replies.

This is an interesting paper. it is not clear if the first surgical indication was the infertility or the abnormal semen analysis. another question is the semen improvement after surgery still remain under "normal" values or had similar abnormalities. 

Reply : Thank you very much for your comments. The first indication for embolization was abnormal semen analysis in patients with stage 3 varicocele in a context of couple infertility. It has been clarified in the « patients » paragraph of the « materials and methods » section for more understanding. Regarding the improvement, among the 94 patients, a significant improvement in the median TSN after VE was shown (31.79x106/ejaculate [IQR:11.10-127.40x106/ejaculate] versus 62.24x106/ejaculate [IQR:17.90-201.60x106/ejaculate], p= 0.0295). Significant improvement in TSN was found for the 60 oligo- or azoospermic patients (p=0.0007) whereas no significant change was found for the 42 patients with normal initial TSN (p=0.49). Other parameters such as progressive and total SMot, SV and SMor also significantly improved after VE (p=0.0003, 0.0013, 0.0356 and 0.007, respectively). Despite still under normal values after embolization, the improvement was statistically significant for most of parameters, the absolute value of improvement being more important than normal values themselves, even if pregnancy rates after embolization was not assessed in our study. Our goal was to show that VE allows semen parameters improvement. It is already described in the limitations section of the manuscript.

Again, had those having a pregnancy immediatly after surgery, a severe abnormal semen before surgery? 

Reply : Thank you very much for your comments. As mentioned before, pregancy rate after embolization was not assessed in our study. The goal was fisrt to show that embolization with NBCA improves the semen parameters in men with couple infertility. We know that couple infertility may be multifactorial (man and woman) and that it is very difficult to evaluate the relationship between improvement of semen parameters and pregnancy rate. It needs further studies with complex analysis that will be done in a second step. However, semen parameters improvement due to embolization is very promising for future couple fertility.

Probably it is interesting to undestand who had the best improvement independently by the procedure used !  

Reply : Thank you very much for your comments. We fully agree. This analysis on improvement according to the type of sperm alteration type (normal TSN, oligospermia and azoospermia) has already been done and compared. Further multivariate analysis was impossible to do given the heterogeneity of the population and the potential number of variables to study. However, among 94 patients, a greater improvement of the TNS was found for the 52 patients who presented initial abnormal TSN than for the 42 with normal initial TSN (median, 133% [IQR:-6–648] versus 4% [IQR: -45–104]). Box plots shown in Figure 3 confirms these results with a significant improvement in the TSN among initially oligo or azoospermic patients (p=0.0007), whereas there was no significant change for patients with normal initial TSN (p=0.49). It is described in paragraph 3.2 of the” results” section.

Due to the fact that there are no hormonal test, it is important to explain during your limits that some of these patients may had hypogonadism, and sometime azoospermia or oligo may not required surgery.

Reply : Thank you very much for your comments. It has been added in the « limitations » section as suggested.